# Rapid Quantitative Method Development for Beef and Pork Lymph Nodes Using BAX^®^ System Real Time *Salmonella* Assay

**DOI:** 10.3390/foods12040822

**Published:** 2023-02-15

**Authors:** David A. Vargas, Gabriela K. Betancourt-Barszcz, Sabrina E. Blandon, Savannah F. Applegate, Mindy M. Brashears, Markus F. Miller, Sara E. Gragg, Marcos X. Sanchez-Plata

**Affiliations:** 1International Center for Food Industry Excellence, Department of Animal and Food Sciences, Texas Tech University, Lubbock, TX 79409, USA; 2HygienaTM, 2 Boulden Circle, New Castle, DE 19720, USA; 3Department of Animal Science & Industry, Kansas State University, Manhattan, KS 66506, USA

**Keywords:** beef lymph nodes, pork lymph nodes, *Salmonella* quantification, pathogen control

## Abstract

The goal of this study was to develop a rapid RT-PCR enumeration method for *Salmonella* in pork and beef lymph nodes (LNs) utilizing BAX^®^-System-SalQuant^®^ as well as to assess the performance of the methodology in comparison with existing ones. For study one: PCR curve development, pork, and beef LNs (n = 64) were trimmed, sterilized, pulverized, spiked with 0.00 to 5.00 Log CFU/LN using *Salmonella* Typhimurium, and then homogenized with BAX-MP media. Samples were incubated at 42 °C and tested at several time points using the BAX^®^-System-RT-PCR Assay for *Salmonella*. Cycle-Threshold values from the BAX^®^-System, for each *Salmonella* concentration were recorded and utilized for statistical analysis. For study two: Method comparison; additional pork and beef LNs (n = 52) were spiked and enumerated by (1) 3M™EB-Petrifilm™ + XLD-replica plate, (2) BAX^®^-System-SalQuant^®^, and (3) MPN. Linear-fit equations for LNs were estimated with recovery times of 6 h and a limit of quantification (LOQ) of 10 CFU/LN. Slopes and intercepts for LNs using BAX^®^-System-SalQuant^®^ when compared with MPN were not significantly different (*p* < 0.05), while the same parameters for 3M™EB-Petrifilm™ + XLD-replica plate were significantly different (*p* > 0.05). The results support the capability of BAX^®^-System-SalQuant^®^ to enumerate *Salmonella* in pork and beef LNs. This development adds support to the use of PCR-based quantification methodologies for pathogen loads in meat products.

## 1. Introduction

Microbial pathogens cause many foodborne illnesses and outbreaks across the world. According to the Centers for Disease Control and Prevention (CDC), *Salmonella* is the second most common cause of single-etiology outbreaks and causes the most outbreak-associated hospitalizations in the United States [1]. Each year *Salmonella* causes more than 1 million foodborne illnesses, of which 6 to 13% of these illnesses are associated with pork and beef products [2,3,4]. This public health issue has come to the attention of the meat and poultry industry which is continuously exploring new approaches to reduce the risk of *Salmonella* contamination in their products [5]. The primary source of *Salmonella* infections from food animal comes from swine, poultry, and cattle, especially if the animal food products are unproperly handled or undercooked [6].

Sanitary dressing procedures and HACCP-based systems implemented by the industry during the slaughter and processing stages are fundamental steps to avoid organ and carcass contamination. The muscle of the animal is usually sterile, but it can become easily contaminated with *Salmonella* during several identified processing steps due to potential contact with manure and contaminated viscera, equipment, employees, and the processing environment [7]. This is why it is of high importance to have fully trained and properly equipped employees in order to reduce cross-contamination during the processing of meat products. Despite having these preventive systems implemented in processing facilities, there is evidence that other sources of contamination can enter the processing stream, including lymph nodes and other glands [7].

Recent studies conducted in commercial settings have shown that the lymphatic system and lymph nodes (LNs), in particular, may be a possible source of *Salmonella* contamination in animal food products, especially in comminuted pork and beef products, due to their obstructed location in the adipose tissue that is usually trimmed and ground with animal muscle [8]. The lymphatic system has an immune function in the body of the animal, working as a filter for bacteria and viruses in order to sequester and inactivate them, thereby ensuring the health of the animal [9,10]. Although mesenteric LNs are discarded during evisceration (i.e., not located in the final product), many other LNs, such as peripheral lymph nodes (PLN: subiliac, superficial cervical, and popliteal), remain situated in the adipose tissue of the carcass and are impossible to be treated by chemical antimicrobial interventions applied during the slaughter and processing stages [8,10,11]. In a cross-sectional cattle study, *Salmonella* was recovered from subiliac LNs of the cull and feedlot cattle, with 33% of contaminated LNs (144) harboring *Salmonella* at 1.9 to >3.8 Log CFU/g, and *Salmonella* prevalence was affected by season, region, and cattle type (cull vs. feedlot) [12]. A different LNs study in pork reported the highest contamination of *Salmonella* in ileocolic LNs (18.8%), ileum (13.9%), mandibular LNs (12.9%), and tonsils (9.9%) of pork carcasses destinated for consumption [13]. These results suggest that if contaminated, these LNs are a direct source of *Salmonella* contamination in the final comminuted product. Removing high-risk LNs during harvest may be favorable to reduce *Salmonella* contamination in products such as comminuted pork and ground beef [8]. However, the high speeds of the line during the slaughter may not give sufficient time to remove all the LNs, and it could be too laborious to do with all the carcasses that are processed during the day, especially in large operations.

In recent years, more importance has been given to the need for enumeration in addition to prevalence in raw meat products, with the understanding that the number of bacteria entering a processing facility is important when assessing the possible risks associated with the harvest and post-harvest stage of swine and beef production [14]. Numerous quantification approaches with limited applicability have been developed, including direct plating using a selective medium, the most probable number technique, among others. However, the need to obtain a less costly and less labor-intensive method has led to the use of PCR for the accurate quantification of pathogens [15]. Different studies have shown that the use of rapid, PCR-based enumeration methods provides a more reliable estimation of pathogen loads at different stages in a processing facility, allowing food safety managers to establish pathogen baselines, estimate statistical process control parameters, evaluate antimicrobial intervention effectiveness, and support decision-making initiatives that would reduce product contamination [16,17].

The goal of this study was to 1) develop a rapid enumeration method for *Salmonella* in pork and beef lymph nodes utilizing the BAX^®^-System-SalQuant^®^ and 2) assess the performance of the newly developed methodology with the existing methodology for quantification of *Salmonella* in lymph nodes using both 3M™ EB Petrifilm™ + XLD replica plate and the standard Most Probable Number (MPN) method conducted through PCR detection for quantification estimation of pathogens in LNs.

## 2. Materials and Methods

### 2.1. Study one: Curve Development

#### 2.1.1. Sample Collection and Processing

Pork lymph nodes (superficial inguinal and subiliac) and beef lymph nodes (mesenteric, superficial inguinal, and subiliac) were collected from pork and beef carcasses at commercial pork and beef processing facilities and then immediately chilled and shipped overnight to the ICFIE Food Microbiology laboratory at Texas Tech University for microbiological analysis (n = 64). Lymph nodes (LN) were trimmed, removing all surrounding fat and fascia, weighed, surface sterilized by immersion in boiling water for 3–5 s, placed into individual sterile bags (Nasco, Atlanta, GA, USA), and pulverized using a rubber mallet. Two SalQuant^®^ quantification curves were developed for each species: one curve for small lymph nodes (pork < 3 g and beef < 10 g) and one curve for medium lymph nodes (pork ≥ 3 g and ≤ 25 g; and beef ≥ 10 g but less than 50 g). A total of 32 pork lymph nodes and 32 beef lymph nodes were processed.

#### 2.1.2. Inoculation Procedure

*Salmonella enterica* subsp. *enterica* ser. Typhimurium (ATCC 14028) from the ICFIE culture collection was removed from the −80 °C freezer in a solution of 10% glycerol, and a 10 µL loop was streaked to brain heart infusion (BHI) agar plates (Millipore Sigma, Danvers, MA, USA) in triplicate for later incubation at 37 °C for 24 h. After incubation, a well-isolated colony was placed into a 9 mL BHI broth (Millipore Sigma, Danvers, MA, USA) tube in triplicate and incubated at 37 °C for 24 h. Following incubation, 0.1 mL of the culture was added to 9 mL of BHI broth (Millipore Sigma, Danvers, MA, USA), followed by incubation at 37 °C for 18 h to obtain a concentration of 9.0 Log CFU/mL. The incubated cultures were serially diluted in buffered peptone water (BPW) tubes (Millipore Sigma, Danvers, MA, USA), plated onto BHI agar in duplicate (Millipore Sigma, Danvers, MA, USA), and incubated at 37 °C for 24 h. Colonies were counted and reported as log_10_ CFU/mL. 

According to their weight, 16 pork lymph nodes were enriched with 20 mL (<3 g) and 16 pork lymph nodes with 80 mL (≥3 g and ≤25 g) of BAX MP (Hygiena, Camarillo CA, USA), and homogenized using a bag mixer (Model 400 circulator, Seward, West Sussex, UK) at 230 rpm for 1 min (lymph node homogenate). The same process was followed for beef lymph nodes, but 16 beef lymph nodes were enriched with 40 mL (<10 g) and 16 beef lymph nodes with 80 mL (≥10 g and ≤50 g) of BAX MP (Hygiena, Camarillo CA, USA). After homogenization, an aliquot of 3 mL was extracted from the lymph node homogenates (LNH), placed into sterile tubes, and incubated at 42 °C for 24 h. The bags with the remaining LNH were stored at 4 °C for 24 h. After incubation of the 3 mL aliquot of LNH, the BAX^®^ System RT-PCR Assay for *Salmonella* (Hygiena, Camarillo, CA, USA) was followed to pre-screen samples for potential pathogen presence. Samples positive for *Salmonella* were discarded, while the negative samples were reserved for later inoculation for this study.

#### 2.1.3. Microbiological Analysis

Using the *Salmonella* negative LNH bags, three biological replicates were inoculated per *Salmonella* concentration (1 to 5 Log CFU/LN), plus a negative control that was not inoculated for each of the two curves and species. All samples were incubated at 42 °C and tested at 6, 8, and 24 h. At each time point, samples were removed from the incubator and analyzed in quintuplets using the BAX^®^ System RT-*Salmonella* Assay (Hygiena, Camarillo, CA, USA). Cycle Threshold values from the BAX^®^ System associated with each pathogen concentration were recorded and utilized for statistical analysis for pathogen load estimation. 

#### 2.1.4. Statistical Analysis

All data were analyzed using R (Version 4.1.3) statistical analysis software to evaluate the relationship between the Cycle Threshold (CT) values obtained from the BAX^®^ System and the inoculation levels. Counts were transformed into Log CFU/LN, and a linear model was calculated for each timepoint where log_10_ counts from each inoculation level were considered as the independent variable, while CT values were considered as the dependent variable. R-squared values and Log root mean squared errors (Log RMSE) were calculated to define the best regression curve. 

### 2.2. Study Two: Method Comparison and Verification

#### 2.2.1. Sample Collection and Processing

Superficial inguinal pork lymph nodes and subiliac beef lymph nodes were collected from pork and beef carcasses at commercial pork and beef processing facilities and then immediately chilled and shipped overnight to the ICFIE Food Microbiology laboratory at Texas Tech University for microbiological analysis (n = 54). Lymph nodes were trimmed, removing all surrounding fat and fascia, weighed, surface sterilized by immersion in boiling water for 3–5 s, placed into individual sterile bags (Nasco, Atlanta, GA, USA), and pulverized using a rubber mallet. A total of 26 pork lymph nodes and 26 beef lymph nodes were processed.

#### 2.2.2. Inoculation Procedure

According to their weight, 26 pulverized pork lymph nodes were enriched with 20 mL (<3 g) or 80 mL (≥3 g and ≤25 g) of BAX MP (Hygiena, Camarillo CA, USA) and homogenized using a bag mixer (Model 400 circulator, Seward, West Sussex, UK) at 230 rpm for 1 min (lymph node homogenate). The same process was followed for beef lymph nodes but enriched with 40 mL (<10 g) or 80 mL (≥10 g and ≤50 g) of BAX MP (Hygiena, Camarillo, CA, USA). After homogenization, an aliquot of 3 mL was extracted from LNHs, placed into sterile tubes, and incubated at 42 °C for 24 h. The bags with the remaining LNH were stored at 4 °C for 24 h. After incubation, the BAX^®^ System RT-*Salmonella* Assay (Millipore Sigma, Danvers, MA, USA) was followed to pre-screen samples. Samples positive for *Salmonella* were discarded, while the negative samples were reserved for later inoculation.

*Salmonella enterica* subsp. *enterica* ser. Typhimurium (ATCC 14028) was used for inoculation. An isolate was obtained from a −80 °C freezer in a solution with 10% glycerol and streaked into brain heart infusion (BHI) agar plates (Millipore Sigma, Danvers, MA, USA) for later incubation at 37 °C for 24 h. A well-isolated colony from the BHI agar plate was suspended in 9 mL of BHI broth (Millipore Sigma, Danvers, MA, USA) and incubated at 37 °C for 24 h. Then, 0.1 mL of the culture was added to 9 mL of BHI broth (Millipore Sigma, Danvers, MA, USA), but this time incubated at 37 °C for 18 h to obtain a concentration of 9.0 Log CFU/mL. The final cocktail was serially diluted in BPW tubes (Millipore Sigma, Danvers, MA, USA). Using the *Salmonella* negative LNH bags, five samples were spiked with 1.00 to 5.00 Log CFU/LN plus a negative control that was not inoculated, resulting in a total of 26 samples each for pork and beef lymph nodes.

#### 2.2.3. Pathogen Enumeration

From each inoculated LNH, samples were enumerated by (1) a direct-plating method on Enterobacteriaceae (EB) 3M™ Petrifilms^TM^ that were replica-plated onto selective XLD plates, (2) a 3 × 3 most probable number (MPN) conducted using the BAX^®^ System RT-*Salmonella* Assay, and (3) the BAX^®^-System-SalQuant^®^ methodology for pork and beef lymph nodes.

(1)3M™ EB Petrifilm™ + XLD replica plate: from each sample, the Association of Official Agricultural Chemists 2003.01 (AOAC) method was used [18]. A 1 mL aliquot of the LNH was plated in Enterobacteriaceae petrifilms (3M, Saint Paul, MN, USA) and incubated at 37 °C for 24 h. After incubation, Enterobacteriaceae counts on petrifilms were recorded. Next, a replica plate from the petrifilm was prepared using xylose lysine deoxycholate (XLD; Remel, San Diego, CA, USA) by placing the film containing the inoculated petrifilm agar on top of the XLD plate and gently pushing for colony transfer. The replica-plated XLD selective plates were incubated for 16 h at 37 °C, and *Salmonella* counts on plates were collected.(2)3 × 3 MPN+ BAX^®^ System RT-*Salmonella* Assay: the microbiological laboratory guidelines (MLG) Appendix 2.05 published by the United States Department of Agriculture (USDA) was used as a guideline for performing this method [19]. A 3 BPW (Millipore Sigma, Danvers, MA, USA) tube, 3 dilution MPN (3 × 3) was prepared according to the respective countable range for each of the different samples. Tubes were incubated at 37 °C for 24 h, and then the BAX^®^ System RT-PCR Assay for *Salmonella* (Hygiena, Camarillo, CA, USA) was followed to confirm the presence of *Salmonella* in each of the MPN tubes. The subsequent pattern of positive and negative tubes was used to determine the MPN count from MPN tables [19].(3)BAX^®^-System-SalQuant^®^ from the LNH samples were immediately incubated at 42 °C for 6 h for recovery. The 6 h recovery period was determined to be the most accurate based on the curve development procedures outlined in objective 1 (highlighted below in Section 3.1). After recovery, the BAX^®^-System-SalQuant^®^ (Hygiena, Camarillo, CA, USA) methodology was followed for the enumeration of *Salmonella*. Samples were then incubated for 18 h at 42 °C for enrichment. Samples that were not positive for enumeration using BAX^®^-System-SalQuant^®^ (Hygiena, Camarillo, CA, USA) were tested for prevalence analysis (detection) using the BAX^®^ System RT-PCR Assay for *Salmonella* (Hygiena, Camarillo, CA, USA) after 24 h of incubation at 42 °C.

#### 2.2.4. Statistical Analysis

All data were analyzed using R (Version 4.1.3) statistical analysis software to evaluate the relationship between the use of the two alternative enumeration technologies with the standard method. Counts were transformed into Log CFU/LN, and a linear model was calculated for each alternative enumeration method (3M™ EB Petrifilm™ + XLD replica plate and BAX^®^-System-SalQuant^®^) where log_10_ counts from each of the alternative enumeration methods were considered as the dependent variable, while log_10_ counts obtained from 3 × 3 MPN (standard method) were considered as the independent variable.

## 3. Results and Discussion

### 3.1. Curve Development

Linear-fit equation estimations for small and medium pork lymph nodes were created to estimate the best recovery for *Salmonella* quantification (Figure 1). CT values obtained from each timepoint, and each pork lymph node size were used as the dependent variable for the linear regression, while the expected inoculation level (Log CFU/LN) was used as the independent variable. The coefficient of determination (R^2^) is a statistical measure in a regression model that determines the proportion of variance in the dependent variable (CT Value) that can be explained by the independent variable (inoculation level). In other words, the R^2^ indicates how well the data fits the regression model. For this experiment, the R^2^ at 6 h and 8 h of recovery time for small pork lymph nodes were 0.90 and 0.65, respectively (Figure 1). Conversely, the R^2^ at 6 h and 8 h for medium pork lymph nodes were 0.90 and 0.62, respectively (Figure 1).

The root means squared error (RMSE) in linear regression is the standard deviation of the residuals, or how to spread out these residuals from the line of best fit. For this case, the Log RMSE is presented as it provides more information about the variability that may exist while using the CT Values for prediction with new sets of data. For this experiment, the Log RMSE at 6 h and 8 h of recovery time for small pork lymph nodes were 0.469 and 1.037, respectively (Figure 1). Conversely, the Log RMSE at 6 h and 8 h for medium pork lymph nodes were 0.432 and 1.108, respectively (Figure 1).

Similarly, linear-fit equation estimations for small and medium lymph nodes were created to estimate the best recovery time for *Salmonella* quantification in beef lymph nodes. Moreover, R^2^ and Log RMSE values were estimated as measurements of fitness and variability created by the model. The R^2^ at 6 h and 8 h of recovery time for small beef lymph nodes were 0.91 and 0.86, respectively (Figure 2). Conversely, the R^2^ at 6 h and 8 h for medium beef lymph nodes were 0.82 and 0.77, respectively (Figure 2). Furthermore, the Log RMSE values at 6 h and 8 h of recovery time for small beef lymph nodes were 0.440 and 0.572, respectively (Figure 2), while the Log RMSE at 6 h and 8 h for medium beef lymph nodes were 0.628 and 0.752, respectively (Figure 2).

The MPN method is laborious and time-consuming, and results are obtained after 24 h [20,21]. When enumerating high concentrations, the MPN method tends to overestimate the count; however, when used at low concentrations, the MPN method is very precise and accurate, which is why the most probable number method has been used by the United States Department of Agriculture (USDA) to enumerate low concentrations of *Salmonella* in food matrices [20,21,22]. However, the need for more rapid, affordable, and reliable methods of enumeration for *Salmonella* is crucial because of the increasing concerns about the safety of food products and regulatory changes by government authorities. In recent years, indicator organisms have been tested in the meat and poultry industry to evaluate the performance of food safety management systems in a processing facility [23,24,25,26,27]. The Enterobacteriaceae family is the indicator organism usually selected for this analysis because *Salmonella* and *Escherichia coli* are part of this group of bacteria. Even though these pathogenic bacteria belong to this bacterial family, several studies suggest that the correlation between *Salmonella* concentrations and Enterobacteriaceae concentrations is not high, indicating that the best indicator for pathogen load will always be the enumeration of the pathogen itself [23,28,29]. The BAX^®^-System-SalQuant^®^ has already been tested in multiple other studies with great performance, including in multiple species and with a wide variety of sample types [16,17,29].

### 3.2. Method Comparison and Verification

For the methodology comparison experiment, counts were log_10_ transformed and then analyzed. Simple linear regression analysis estimates two parameters that establish the relationship between two variables and can be used to estimate an average rate of change. The slope of the linear model represents the estimated increase in the dependent variable per unit increase in the independent variable. The greater the magnitude of the slope, the greater the rate of change between the variables. In this case, the slope represents the average rate of change in microbial counts using the 3M™ EB Petrifilm™ + XLD replica plate or BAX^®^-System-SalQuant^®^ methodologies due to an incremental change of 1 unit in the MPN method in pork lymph nodes. Ideally, the slope should take a value of 1, suggesting that for any 1 Log CFU/LN increase using any of the alternative methodologies, there is an increase in 1 Log CFU/LN by using the standard methodology (Figure 3). The slope for the 3M™ EB Petrifilm™ + XLD replica plate method was 1.248 with an adjusted r-squared of 0.93 and a 95% confidence interval from 1.100 to 1.395, while the slope for the BAX^®^-System-SalQuant^®^ method was 1.023 with an adjusted r-squared of 0.75 and a 95% confidence interval from 0.770 to 1.275 (Table 1).

Moreover, the intercept represents the value associated with the dependent variable when the independent variable is equal to zero. For this case, the intercept represents the value measured by using the alternative method when the MPN method is associated with a value of zero, representing the similarity in the total magnitude read by both methods (Figure 3). The intercept value for the 3M™ EB Petrifilm™ + XLD replica plate method was −1.003 with a 95% confidence interval from −1.501 to −0.505, while the intercept for the BAX^®^-System-SalQuant^®^ method was −0.333 with a 95% confidence interval from −1.186 to 0.519 in pork lymph nodes (Table 1). 

The estimated slope for the BAX^®^-System-SalQuant^®^ was 1.023, suggesting that for 1 Log CFU/LN increase in counts using the standard method, a 1.023 Log CFU/LN average increase in counts will be obtained using the BAX^®^-System-SalQuant^®^ method. The 95% confidence interval of the slope was estimated to evaluate if there is no statistical difference between the estimated slope and the value of 1, as no differences suggest that the estimation of the BAX^®^-System-SalQuant^®^ is comparable to the modified MPN method. More specifically, the 95% confidence interval contained the value of 1, suggesting that the differences observed were due to random chance or random experimental error. Moreover, the similarity in the total magnitude, measured by the intercept of the model, indicates the estimation of the BAX^®^-System-SalQuant^®^ method when the MPN method is associated with a value of 0 Log CFU/LN. When performing linear regression analysis, the *p*-value for the intercept is calculated by evaluating the null hypothesis that the intercept is equal to zero. The obtained *p*-value for this model was 0.427, which provides enough evidence that the intercept was not different from zero; thus, on average, the value obtained by using the BAX^®^-System-SalQuant^®^ is 0 Log CFU/LN when the MPN method counts 0 Log CFU/LN. The graphical representation of this relationship (Figure 3) illustrates greater variability in the results at higher inoculation levels, which is easily explained by the confidence intervals obtained during the development of the curve (Figure 1). However, on average, the results suggest that BAX^®^-System-SalQuant^®^ is very effective at estimating *Salmonella* concentrations in LN samples when compared with the standard method.

Furthermore, the same analysis was performed with the 3M™ EB Petrifilm™ + XLD replica plate method, as this method has been used in prior studies for the quantification of *Salmonella* in lymph nodes [12]. The slope was 1.248, suggesting that for 1 Log CFU/LN increase in counts using the standard method, a 1.248 Log CFU/LN average increase in counts will be obtained using the 3M™ EB Petrifilm™ + XLD replica plate method (Table 1). The 95% confidence interval did not contain the value of 1, and the intercept was statistically different from zero, indicating that there are differences in *Salmonella* enumeration between both methodologies (Table 1). The graphical representation of the relationship between the 3M™ EB Petrifilm™ + XLD replica plate method and MPN clearly demonstrates that this alternative methodology is less effective at detecting values lower than 2 Log CFU/LN (Figure 3). This issue can be easily explained by the limit of quantification of the 3M™ EB Petrifilm™, as the lower value that can be estimated is 1 CFU/mL, which is equivalent to 20 CFU/LN (the case where the smaller amount of liquid media was added, without considering the actual weight of the lymph node). The corresponding log_10_ limit of quantification is 1.30 Log CFU/LN. Essentially, the 3M™ EB Petrifilm™ + XLD replica plate quantification method will be less effective at detecting *Salmonella* in larger lymph nodes. In contrast, at higher concentrations, the precision and accuracy of the method are very high when compared with the standard MPN method (Figure 3). 

Multiple studies have evaluated the presence of *Salmonella* in pork lymph nodes. Chaves et al. [30] reported prevalence values of 91.2% in mesenteric lymph nodes, 76.5% in subiliac lymph nodes, 55.9% in mandibular lymph nodes, and 18% in tonsils from swine slaughtered in Mexico. Kampelmacher et al. [31] reported a 15% *Salmonella* prevalence in mesenteric lymph nodes in market hogs. Viera-Pinto et al. [13] reported 18.8% and 9.9% in ileocolic lymph nodes and tonsils, respectively, in swine, and recently Miller et al. [32] reported an overall *Salmonella* prevalence of 21.8% (43/197) in mesenteric, inguinal, subiliac, and tracheobronchial lymph nodes of market hog carcasses at slaughter, suggesting that *Salmonella* can be harbored in multiple lymph nodes in swine. The observed variability of *Salmonella* prevalence in swine identifies an important research opportunity for lymph node mapping in pork carcasses, not only based on prevalence, but also on levels of contamination. This information will allow processors to conduct better informed risk-based decisions about the removal of lymph nodes, and access to rapid and reliable quantification methodologies will be critical for timely decision-making.

Likewise, slope and intercept parameters were estimated for beef lymph nodes (Figure 4). The slope for the 3M™ EB Petrifilm™ + XLD replica plate method was 1.187 with an adjusted r-squared of 0.84 and a 95% confidence interval from 0.965 to 1.409, while the slope for the BAX^®^-System-SalQuant^®^ method was 1.092 with an adjusted r-squared of 0.83 and a 95% confidence interval from 0.879 to 1.305 (Table 2). The intercept value for the 3M™ EB Petrifilm™ + XLD replica plate method was −1.064 with a 95% confidence interval from −1.836 to −0.293, while the intercept for the BAX^®^-System-SalQuant^®^ method was −0.251 with a 95% confidence interval from −0.992 to 0.489 in beef lymph nodes (Table 2). 

Similar interpretations and results were obtained for beef lymph nodes for both methodologies. The estimated parameters for the BAX^®^-System-SalQuant^®^ indicated that there were no differences when compared with the MPN method. Conversely, both the slope and intercept for the 3M™ EB Petrifilm™ + XLD replica plate method were significantly different when compared with the standard MPN method. Moreover, similar problems with a limit of quantification were identified for beef lymph nodes, as beef lymph nodes tend to be even larger than pork lymph nodes. However, on average, both alternative methodologies are successful at estimating results at higher microbial concentrations (Figure 4).

There are also multiple studies evaluating the prevalence and concentrations of *Salmonella* in beef lymph nodes. Samuel et al. [33] reported 72% (61/85) of cattle in the study harbored *Salmonella* in lymph nodes (jejunal or cecal), with concentrations of *Salmonella* estimated at up to 5000 CFU/g in mesenteric lymph nodes. Another study that detected and enumerated *Salmonella* in chuck and flank lymph nodes reported an overall prevalence of 1.6%, with one flank node from a cull cow contaminated with 5.8 CFU/g of *Salmonella*. Gragg et al. [12] reported a median point estimate of *Salmonella enterica* of 1.3% in beef subiliac lymph nodes, 67% of which harbored <0.1 to 1.8 Log CFU/g, while 33% ranged from 1.9 to >3.8 Log CFU/g. All studies mentioned have different methodologies for the enumeration of *Salmonella,* all of which could be considered time-consuming and expensive, with results obtained in more than 24 h. Recently, the United States Department of Agriculture Food Safety and Inspection Service took steps to reduce *Salmonella* in meat products with the release of new performance standards for pork products, a new proposed regulatory framework to reduce *Salmonella* infections linked to poultry products, as well as an announcement to declare *Salmonella* as an adulterant in breaded stuffed raw chicken products [34,35]. These regulatory actions emphasize enumeration, which evidently shows the importance of *Salmonella* enumeration in meat products to government authorities in the United States, as well as the need for development of methodologies that are accurate, precise, with low limits of quantification and detection, fast, and reliable with results [34].

## 4. Conclusions

The results of these studies validated the ability of BAX^®^-System-SalQuant^®^ to enumerate *Salmonella* in small and medium pork and beef lymph nodes. The method adequately estimated all concentrations of *Salmonella* when compared with the standard MPN method from 10 to 100,000 CFU/LN within 6 h. This development and validation study provides a rapid and feasible quantification methodology that can serve as a tool for the meat industry when conducting contamination risk assessments and adds support to the use of PCR-based quantification methodologies as a food safety management tool for ongoing monitoring of pathogen contamination and risk-based product disposition. Future research should address the use of the methodology in real commercial facilities and create a baseline about the levels of *Salmonella* in different pork and beef lymph nodes.

## Figures and Tables

**Figure 1 foods-12-00822-f001:**
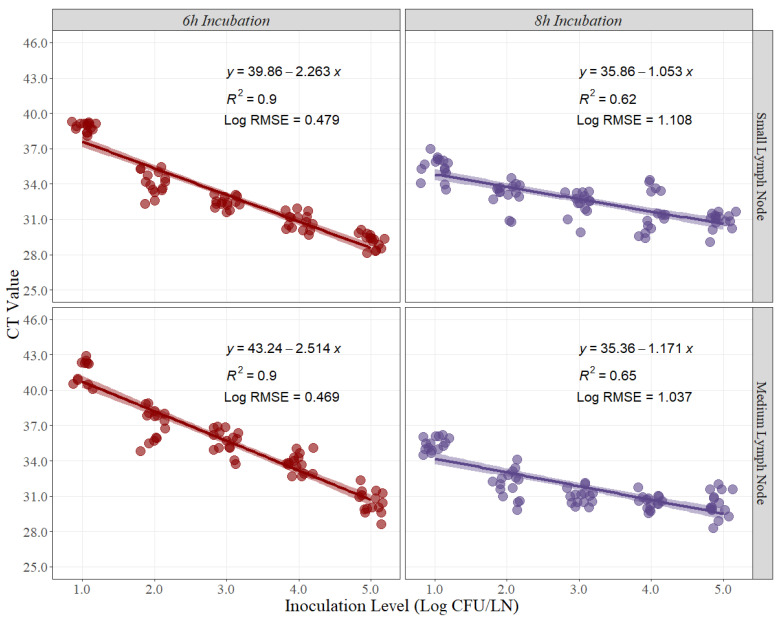
Linear regression for curve development of small and medium pork lymph nodes at 6 h and 8 h of recovery time, comparing cycle threshold (CT) values and inoculation levels (Log CFU/LN), (n = 15 per inoculation level). A small level of jittering was added to the actual dots for visualization purposes. RMSE = root mean squared error.

**Figure 2 foods-12-00822-f002:**
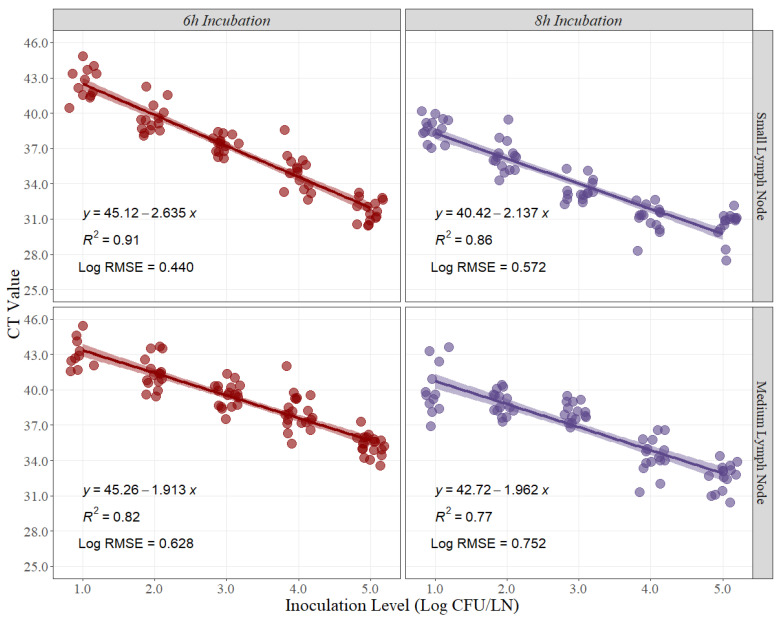
Linear regression for curve development of small and medium beef lymph nodes at 6 h and 8 h of recovery time, comparing cycle threshold (CT) values and inoculation levels (Log CFU/LN), (n = 15 per inoculation level). A small level of jittering was added to the actual dots for visualization purposes. RMSE = root mean squared error.

**Figure 3 foods-12-00822-f003:**
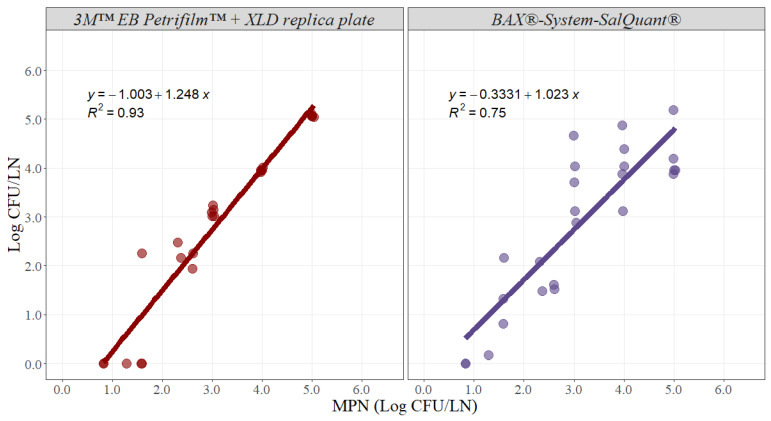
Graphical representation of the linear correlation comparing 3M™ EB Petrifilm™ + XLD replica plate and BAX^®^-System-SalQuant^®^ versus most probably number (MPN) for quantification of *Salmonella* in pork lymph nodes (n = 25 per method). The dots represent the actual data points.

**Figure 4 foods-12-00822-f004:**
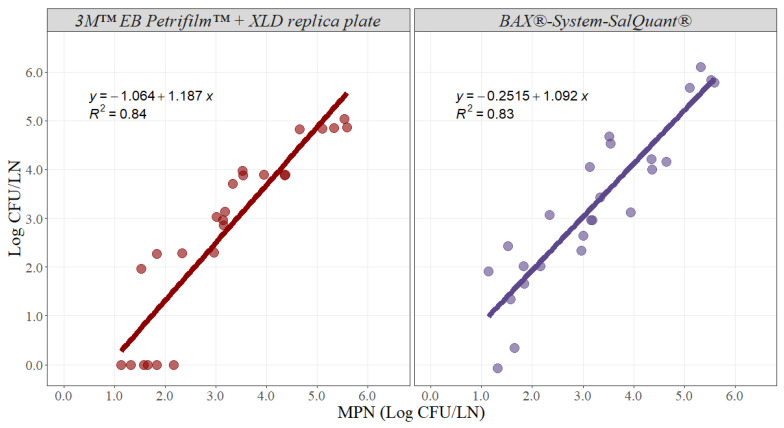
Graphical representation of the linear correlation comparing EB Petrifilm + XLD replica plate and BAX^®^-System-SalQuant^®^ versus MPN for quantification of *Salmonella* in beef lymph nodes (n = 25 per method). The dots represent the actual data points.

**Table 1 foods-12-00822-t001:** Summary table of linear models using the least squares regression method predicting the bacterial counts of 3M™ EB Petrifilm™ + XLD replica plate and BAX^®^-System-SalQuant^®^ when compared with MPN for quantification of *Salmonella* in pork lymph nodes (n = 25 per method).

Enumeration Method	Coefficient	Estimate	Standard Error	*p*-Value	95% Confidence Intervals	
					Lower (2.5%)	Upper (97.5%)
3M™ EB Petrifilm™ + XLD replica plate	Intercept	−1.003	0.241	<0.001	−1.501	−0.505
	Slope	1.248	0.071	<0.001	1.100	1.395
BAX^®^-System-SalQuant^®^	Intercept	−0.333	0.412	0.427	−1.186	0.519
	Slope	1.023	0.122	<0.001	0.770	1.275

**Table 2 foods-12-00822-t002:** Summary table of linear models using the least squares regression method predicting the bacterial counts of 3M™ EB Petrifilm™ + XLD replica plate and BAX^®^-System-SalQuant^®^ when compared with MPN for quantification of *Salmonella* in beef lymph nodes (n = 25 per method).

Enumeration Method	Coefficient	Estimate	Standard Error	*p*-Value	95% Confidence Intervals	
					Lower (2.5%)	Upper (97.5%)
3M™ EB Petrifilm™ + XLD replica plate	Intercept	−1.064	0.373	0.009	−1.836	−0.293
	Slope	1.187	0.107	<0.001	0.965	1.409
BAX^®^-System-SalQuant^®^	Intercept	−0.251	0.358	0.489	−0.992	0.489
	Slope	1.092	0.103	<0.001	0.879	1.305

## Data Availability

Data is available on request from the corresponding author.

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
