# Peer review of "Rapid Quantitative Method Development for Beef and Pork Lymph Nodes Using BAX® System Real Time Salmonella Assay"

_foods, 2023, doi:10.3390/foods12040822_

Round 1
Reviewer 1 Report
See attached detailed recommendations for presentation of results.

Author Response
Reply attached in file.

Reviewer 2 Report
The manuscript deals with the development of a tool consisting in a Rapid Quantitative Method for Beef and Pork Lymph Nodes using BAX® System Real Time Salmonella Assay. The work has a clear objective of contrasting the efficacy of the proposed methodology and achieves interesting results to be evaluated by the scientific community, and with potential application in its sector.
The work is well designed and developed, although we suggest some aspects that would improve and strengthen.
Here are my specific comments on:
Abstract:
Lines 22-23: “using the” is repeated
Introduction
Lines 55-57: It would be convenient to enhance this sentence with a reference
Material and Methods:
Section 2.1.3. is missing. ¿Microbiological analyses?
Line 155: According to the Journal Instruction for Authors, “CT” must be defined the first time is mentioned.
Lines 164-170: Why number of samples are different between PCR curve development and comparison of methods?
Lines 164-170 and section 2.1.4.: It would be convenient to hold with references/s that number of samples is enough to develop a curve for a linear model R.
Line 206: A space between words “and” and “incubated” must be adjusted.
Results and discussion
Lines 273-290: This paragraph seems to fit better as part of introduction. For discussion section, it should be better to compare results to other similar works (if they have been carried out), or to highlight the contribution of this study
Line 318: “For this case” appears as a separated sentence. Is right? I don’t realize what’s the meaning.
References
Line 533: Plata must be written in Capital letter
Author Response
Reply attaached in file.
